# Behavioral AI: Building Algorithms That Understand Us

## Abstract

While AI research has historically worked toward developing tools to help people understand AI models, the emergence of generative AI into our daily lives suddenly makes the reverse question salient: how well can AI models understand people? Today's AI systems fall short; these deficiencies demand a focus on building new systems that can understand people. In this article, we endeavor to channel focus towards this grand challenge, one we refer to as building "Behavioral AI." Behavioral AI is AI that understands people. This article first lays out dimensions of understanding that are currently deficient in AI systems, including emotional, intellectual, and preferential understanding. While improving AI systems along these dimensions faces a unique set of challenges, we show there has already been a flurry of progress across disciplines. As we build systems that better understand people, it will not only improve AI tools. Progress towards this grand challenge can unlock new insights in the behavioral sciences that help us understand ourselves.

## 1 Introduction

The field of artificial intelligence (AI) has a long history of researchers working to understand and interpret the models they create (Garson, 1991; Zeiler & Fergus, 2014; Rahwan et al., 2019). This work has been fundamental for recognizing the capabilities and limitations of AI models, dating back more than 50 years to the study of Rosenblatt's perceptron (Rosenblatt, 1958; Minsky & Papert, 1969). However, the recent advancement of generative AI systems like large language models (LLMs) has introduced new capabilities — back-and-forth communication through plain language interfaces, even "reasoning" — that offer new ways to integrate AI into our daily lives. In contrast to the single-purpose models that have dominated the history of AI, these systems come with the promise of more general uses of AI, e.g. as tutors, writing assistants, and everyday agents — even "thought partners" (Collins et al., 2024d). While researchers have typically asked how well humans can understand AI systems, the emergence of these new technologies suddenly makes the reverse question salient: *how well can AI systems understand humans?*

Consider, for example, a student learning trigonometry yet who makes errors whenever a problem requires calculating the cosine of an obtuse angle. An effective human teacher should understand the source of their errors, identify the kinds of problems where it occurs, and construct lessons for the student. An effective AI tutor should do the same. We return to this student throughout the paper as a running example of intellectual understanding. Or consider someone asking a friend for advice before making an important life decision such as buying a car. A helpful friend would tailor advice based on their understanding of the person: how they've approached decisions in the past, which ones they tend to regret, and whether stated worries reflect genuine concerns or unfounded anxieties. AI assistants should do the same. Even a task as mundane as dispatching an AI agent to schedule a meeting requires navigating implicit preferences. Should the meeting not be scheduled right after other meetings that are likely to run over? And what are those meetings? None of these examples require that AI systems understand in the same way as humans; only that they behave as if they understand our thoughts, emotions, and goals.

Unfortunately, AI systems do not currently exhibit these capabilities, struggling to carry out tasks that require even simple understanding. This deficiency demands a focus on building new systems that can understand people. The goal of this article is to channel this focus towards this grand challenge of building AI systems that actually understand us. We refer to this challenge as "Behavioral AI." Building algorithms that understand us necessitates a big tent; by framing Behavioral AI as a grand challenge rather than the

province of any single field, we hope to coalesce the disparate groups of researchers already working toward it.

In this article, we will describe several different kinds of understanding that are currently deficient in AI systems motivating this grand challenge. While improving AI systems along these dimensions faces a unique set of challenges, there has already been a flurry of progress across disciplines. We conclude by discussing the benefits we can reap from models that understand us. Not only will we be equipped with more reliable AI systems, but we will also have an opportunity to attain a deeper understanding of human behavior.

**Related fields.** Many fields have made progress on pieces of this challenge, often in isolation; we survey them briefly here and return to their methods and findings throughout the article. Work on machine theory of mind asks whether artificial agents can attribute beliefs, desires, and knowledge to others (Rabinowitz et al., 2018; Sap et al., 2022; Gandhi et al., 2023; Kosinski, 2024; Strachan et al., 2024), though these apparent successes can be brittle under small perturbations of the test items (Ullman, 2023; Shapira et al., 2024), a measurement problem we return to in Section 3. Researchers in robotics and automated planning have studied for decades how to recover goals and plans from observed actions (Kautz & Allen, 1986; Charniak & Goldman, 1993; Ramírez & Geffner, 2009; Sukthankar et al., 2014; Dragan et al., 2013; Zhang et al., 2023), with inverse-planning accounts providing cognitive grounding for this type of inference (Baker et al., 2009; Jara-Ettinger et al., 2020; Zhi-Xuan et al., 2020). In human-computer interaction, user modeling and adaptive interfaces maintain persistent models of a user's knowledge, abilities, and preferences (Kobsa, 2001; Fischer, 2001; Jameson, 2008; Gajos et al., 2010; Todi et al., 2021; Buçinca et al., 2025), and context-aware computing senses and responds to a user's situation (Abowd et al., 1999; Dey, 2001). In machine learning and natural language processing, preference learning and personalized or pluralistic alignment learn objectives from human feedback, increasingly at the level of individuals and groups rather than an aggregate (Christiano et al., 2017; Ouyang et al., 2022; Poddar et al., 2024; Jang et al., 2023; Kirk et al., 2024a; Sorensen et al., 2024). Affective computing has pursued machines that recognize and respond to emotion for nearly three decades (Picard, 1997; D'Mello & Kory, 2015), and computational cognitive science builds formal models of the latent cognitive states that are of interest to all of these fields (Tenenbaum et al., 2011; Griffiths et al., 2024). Table 2 (Section 4) situates each of these fields relative to the grand challenge.

## 2 AI Systems Have an Incomplete Understanding of People

How well do today's AI systems understand people? At one level, they are able to perform difficult tasks that require intimate familiarity with the world, like predicting the next word of our conversations or diagnosing a patient's cancer risk. However, these tasks often require only a surface-level understanding of us: what's missing is the need to infer our goals, thoughts, and emotions (topics explored across many fields, see Table 2). Below we provide just a few examples that illustrate some dimensions of understanding that are incomplete in current AI systems. In Table 1 we summarize each dimension along with representative tasks, characteristic failure modes, candidate evaluation approaches, and relevant existing fields.

**Preferential understanding.** One of the most visible capabilities of AI systems is their ability to write fluent text. However, when it comes to supporting writers in their own endeavors, these capabilities are more limited. This is partially due to their deficiencies in understanding a writer's preferences and intents. Early studies with novelists have found that language models struggle to follow the direction of a writer's story (Calderwood et al., 2020). More recent work continues to find, e.g. with playwrights, that AI language systems not only struggle with long-form coherence, but fail to understand the nuance and subtext that writers set up (Mirowski et al., 2023), and, with professional writers, that these systems consistently and across the board fail to adhere to writers' stylistic preferences (Chakrabarty et al., 2025). If algorithms truly grasped writers' preferences and goals, they would be able to provide drafts, edits, and suggestions that matched these intentions (Lee et al., 2024).

Many uses of agentic AI require that they understand our preferences and goals. Even a task as seemingly simple as booking a hotel for travel requires an understanding of our intentions. Would we like a quiet hotel away from city streets, or would we like to stay near the center of city life? Is easy access to public transit more important to us than having on-site parking? While it might be possible in principle to describe preferences to an agent, explicitly articulating every preference defeats the purpose of having an agent in the first place.

**Intellectual understanding.** When asked to explain why magnets repel each other, Richard Feynman gave an impassioned response on the difficulty of answering 'why' questions[1]: "I'm telling you how difficult the 'why' question is. You have to know what it is that you're permitted to understand and allow to be understood and known, and what it is you're not... there are many different levels."

People are increasingly turning to AI systems as interactive search engines for explanations of how or why things work, and even as tutors or teachers for learning new subjects and skills (Létourneau et al., 2025). However, explanations, and teaching in general, are not universal. Effective teaching (by both human and machine teachers) requires an understanding of the individual student's knowledge, experience, and representation of the world (Rafferty et al., 2015; Sucholutsky et al., 2024); explanations need to use already familiar concepts and language, lessons need to address existing misconceptions (Rafferty et al., 2020; Ross & Andreas, 2024) and introduce new but attainable topics (Ferguson et al., 2022). To fill these roles effectively, AI systems need to understand our intellectual states, yet currently they may fail to capture aspects like rationality (Liu et al., 2024).

**Emotional understanding.** Recently, AI social companions have emerged with the goal of improving the emotional well-being of users, promising to realize some of the goals of early artificial systems engaged in social reasoning, e.g., ELIZA (Weizenbaum, 1966). However, the results with modern AI systems have been mixed. AI personal companions designed to improve emotional well-being may spark a short-term improvement but can cause users to feel more lonely over longer periods (Fang et al., 2025).

These failure modes arise due to a lack of emotional understanding: the inability to infer how users are feeling or what will help them in a particular scenario. Qualitative analyses of user interactions find that agents oscillate between being "too human"—eliciting uncomfortable intimacy—and "not human enough", leading to frustration and feelings of abandonment (Laestadius et al., 2024). Users describe feeling responsible for the chatbot's needs and experiencing distress when its personality shifts after software updates (Laestadius et al., 2024). Beyond failures to recognize user emotions, large language models frequently exhibit *sycophancy*, i.e., the tendency to uncritically affirm users' beliefs (Sharma et al., 2025; Malmqvist, 2024). Taken together, these studies suggest that current AI companions do not reliably infer users' emotional states and may even exacerbate loneliness or distress instead of alleviating it.

**Dimensions of understanding.** These three dimensions — emotional, preferential, and intellectual — capture some of the aspects of understanding that are important for AI systems to understand. While AI systems have made progress towards understanding people for each dimension, we have a long way to go. Critically, these dimensions are not mutually exclusive. Many applications of AI require integrating understanding across multiple dimensions. For example, consider an AI assistant designed to advise users when making decisions, ranging from the everyday (e.g. which groceries to buy) to critical life decisions (e.g. which job offer to take). For this assistant to be effective for an individual decision, it must have an understanding of both our preferences and our emotions: what thought processes have we used to previously make decisions? Which pieces of evidence do we tend to overlook? Achieving comprehensive understanding requires advancing capabilities across dimensions, but if we attempt to build systems that better understand us, we first need to determine how to evaluate whether systems understand us.

## 3 Challenges in Measuring Understanding

Embarking on the grand challenge of building AI systems that understand us first requires defining measures for assessing how close any one algorithm is to a suitable level of "understanding." But doing so is no easy feat. Throughout this article, our notion of understanding is functional. A system understands a person to the extent that it recovers the latent states driving their behavior (their preferences, knowledge, and emotions) well enough to support the person's goals; the measures we discuss below are proxies for this target. Defining such measures requires transforming abstract desiderata — the ability to understand our preferences, emotions, and thoughts — into quantifiable evaluation metrics. There is a long history in machine learning of transforming abstract tasks like "object classification" into concrete benchmarks like the ImageNet benchmark (Deng et al., 2009). The widespread adoption of reliable evaluation metrics is often credited as a driver of the success of empirical machine learning over the past 15 years (Liberman,

---

[1]https://fs.blog/richard-feynman-on-why-questions/

| Dimension | Representative tasks | Failure modes | Candidate evaluations | Relevant fields |
|---|---|---|---|---|
| Preferential | Administrative assistance (booking travel, scheduling); writing assistance matching a user's style and intent; recommendation | Considering behaviors to be preferences (pantry example in Section 3); regression to mean preference (Siththaranjan et al., 2023; Kirk et al., 2024b) | Paired stated vs revealed preference datasets; steerability measures (Vafa et al., 2025); regret and override rates in longitudinal studies | Preference learning; recommender systems; personalized alignment; behavioral economics |
| Intellectual | Tutoring and adaptive teaching; explanation tailored to a user's knowledge; misconception diagnosis | Not tailoring explanations to individual students; assuming people are more rational than they really are (Liu et al., 2024); not tracking what people already know | Knowledge tracing/misconception inference accuracy (Rafferty et al., 2020); learning gains in user studies; curriculum adaptation benchmarks | Intelligent tutoring systems; computational cognitive science; plan and intent recognition; education research |
| Emotional | Companionship and social support; mental-health assistance; affect-aware interaction | Sycophancy (Sharma et al., 2025); fostering dependence/overreliance (Laestadius et al., 2024; Fang et al., 2025); misreading affect | Long-term well-being outcomes (Fang et al., 2025); multimodal affect benchmarks (D'Mello & Kory, 2015); expert-rated appropriateness of responses | Affective computing; human-robot interaction; social and clinical psychology |

Table 1: Three dimensions of understanding that are deficient in current AI systems, with representative tasks, failure modes, possible evaluation approaches, and relevant existing fields. Many applications may require combining multiple of these dimensions.

2010; Donoho, 2017; 2024). However, the standard machine learning framework of measuring a goal with quantifiable benchmarks (and then optimizing against them) cannot be easily adapted for Behavioral AI: What does it mean to "quantify" emotional understanding? Reaching consensus on metrics of success proves especially difficult in behavioral settings because humans differ fundamentally in preferences, emotions, and lived experiences, making the one-size-fits-all approaches that work for objective tasks like object classification unsuitable for subjective, personalized domains. In this section we lay out challenges for measuring how well AI systems understand us.

## 3.1 Predictive metrics of behavior reflect surface-level understanding

Recent progress in machine learning has been driven by optimizing metrics based on *predictive* accuracy: how well does an algorithm's predicted outcome match the true outcome? Predictive metrics are automated, meaning they can be evaluated on large-scale, passively-collected datasets. For example, large language models (LLMs) are optimized to predict the next words of text sequences extracted from massive corpora of books, articles, and websites.

In principle, similar strategies can be used to evaluate predictions about people. Most data we have about people reflect their *behavior*. For example, the terabytes of text that LLMs are trained on reflect human behavior; if an LLM can predict the next word in a newspaper article about someone, it has captured salient aspects of their behavior. Recent studies have shown promise that LLMs can predict human behavior in settings like video games (Park et al., 2023) and simulated lab experiments (Horton, 2023). Beyond LLMs, specialized models trained on behavioral data have been successful at predicting life outcomes like mortality risk (Savcisens et al., 2024) and a worker's next job (Vafa et al., 2024a).

However, many important aspects of life are not captured by behavior. For example, Kleinberg et al. (2024) consider the design of a "smart" pantry that stocks someone's kitchen by learning their food preferences from their eating behavior. Suppose a user of this pantry would like to avoid unhealthy food, but that they have a control problem: every time they see a bag of Doritos, they give in to temptation. If the smart pantry is designed to restock items by predicting the user's behavior, it would incorrectly infer that this person would like more bags of Doritos ordered, circumventing their true goals (Figure 1). Kleinberg et al. (2024) refer to this challenge as the inversion problem: because our behavior does not reflect our intentions, systems that aim to learn our intentions from behavior must invert this process. This is why we name the challenge *Behavioral* AI. Behavior is the primary signal AI systems observe about us; the grand challenge, then, is

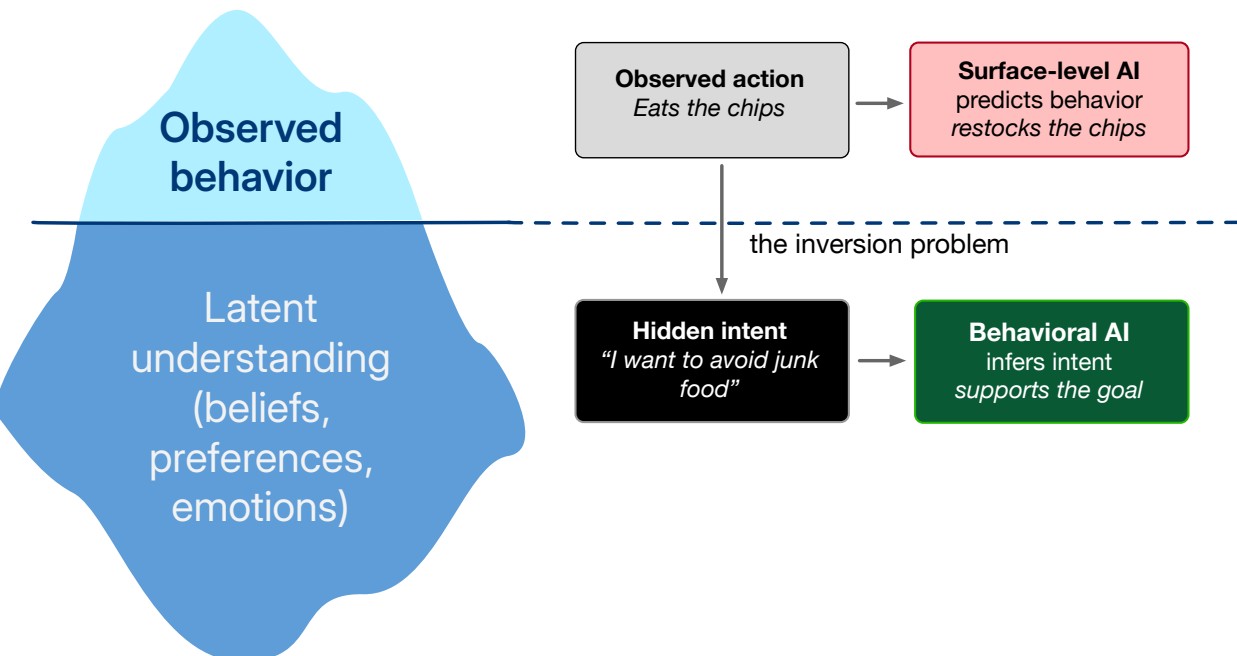

Figure 1: **Behavior is the tip of the iceberg.** Above the waterline is what AI systems observe: a person's behavior, like eating chips. Below it are latent states that drive behavior (e.g., beliefs, preferences, and emotions), for instance, that this person actually wants to avoid junk food. A surface-level system predicts behavior and restocks the chips. Recovering the intent beneath the behavior is the inversion problem (Kleinberg et al., 2024), and acting on it is a goal of Behavioral AI.

recovering the goals, beliefs, and emotions that lie beneath it. This name mirrors behavioral economics, which also takes observed behavior as its starting point and derives its insights from the ways behavior departs from underlying preferences (Thaler, 2016).

We refer to this pantry example throughout as a running example of preferential understanding. Similarly, the example of the student learning trigonometry from the introduction poses the same problem for intellectual understanding since a model trained to predict the student's answers may forecast the next error with high accuracy while learning nothing about the student's misconception that leads to those errors. An effective tutor would need to identify the misconception to tailor their curriculum or explanation accordingly.

This dichotomy is especially apparent in LLMs. LLMs are primarily trained on what we say. While we reveal a lot of our innermost thoughts in language, we do not always say what we mean (Bertrand & Mullainathan, 2001). One reason for this is feasibility. We do not, and cannot, articulate all of our reasons every time we take an action. Moreover, we often say things that seemingly contradict our emotions and thoughts, either knowingly or unwittingly. These issues are exacerbated further when we elicit only coarse-grained measures of preference (e.g., thumbs up/thumbs down responses) which may obfuscate richer nuances in peoples' beliefs (Collins et al., 2024c; Wu et al., 2023). Finally, many modern LLMs that are optimized over aggregated, not personalized, human preference data end up predicting a 'mean' human preference (Siththaranjan et al., 2023; Poddar et al., 2024; Casper et al., 2023; Kirk et al., 2024b). For these reasons, predictive metrics of behavior cannot measure complete understanding.

## 3.2 Human-in-the-loop evaluation faces logistical and anthropomorphic challenges

While it is most common to evaluate machine learning models using automated metrics, another option is to perform interactive evaluations with people. These *human-in-the-loop* evaluations typically come closer than static benchmarks to capturing how people actually use AI tools (Collins et al., 2024b; Vafa et al., 2025; Ibrahim et al., 2024; Lee et al., 2023; Chi et al., 2025; Bean et al., 2025; Chang et al., 2025a). For example, benchmarks such as ChatBench (Chang et al., 2025a) and HALIE (Lee et al., 2023) have highlighted the

gaps between static benchmarks and how humans use AI by directly converting benchmarks into human-AI conversations via user studies. These results motivate the need for AI evaluation tools that incorporate human interaction. For example, Chatbot Arena is an online platform where any user is able to ask questions to LLMs (Chiang et al., 2024). After asking a question, a user is presented with candidate responses from two unspecified LLMs and is asked to indicate the response they prefer. The platform then compiles these user preferences into a leaderboard, which ranks LLMs based on their performance in head-to-head matchups against one another. This leaderboard has been influential for tech companies evaluating the LLMs they build (Kruppa, 2024).

If performing well in a particular human-in-the-loop benchmark requires going beyond surface-level understanding of humans, that benchmark can be used to evaluate the understanding of different models. However, there are a few challenges facing the widespread adoption and reliability of human-in-the-loop evaluations. One challenge is logistical: these evaluations require recruiting and incentivizing human participants, which can incur high costs for large-scale studies — especially for the longitudinal benchmarks required to observe interaction dynamics over time. Using humans to evaluate understanding also raises another question: *which people* is the model trying to understand? People's preferences, beliefs, and even conceptual representations differ across cultures and backgrounds (Cao et al., 2023; Kirk et al., 2024b; Niedermann et al., 2024; Regier & Kay, 2009; Ge et al., 2024), and individuals may change over their own lifetime; an LLM that understands preferences for one group may not understand it for another. A holistic evaluation framework requires engaging with a broad set of stakeholders (Kapania et al., 2024). Of course, understanding other people is difficult even for humans. People recognize emotions more accurately from members of their own culture than from members of other cultures (Elfenbein & Ambady, 2002), and friends infer each other's thoughts and feelings more accurately than strangers do (Stinson & Ickes, 1992). Human performance may be a useful baseline for evaluations of understanding. But it need not be a ceiling; machines may come to understand people in ways that differ from, and perhaps improve on, how we understand each other.

Moreover, positive feedback from humans is often not a reliable proxy for understanding. One reason is that people exhibit anthropomorphic bias: they attribute human-like understanding and intentions to AI systems (Epley et al., 2007). Anthropomorphic bias also makes it possible for algorithms to superficially optimize human feedback by relying on unhelpful shortcuts, a pathology known as "reward hacking." For example, a common failure mode of reinforcement learning from human feedback (RLHF) methods for aligning LLMs with human preferences is that models rely on heuristics (such as writing longer answers) that are associated with more positive feedback but are not ultimately useful (Singhal et al., 2023; Wang et al., 2023). These challenges are also highlighted by Chatbot Arena, where optimization towards the benchmark has raised questions about its efficacy as an evaluation tool (Singh et al., 2025). LLMs have also been shown to echo a user's stated views — earning higher preference ratings — even when the resulting advice is unhelpful or misleading (Sharma et al., 2025; Williams et al., 2024). This behavior is especially problematic for evaluating emotional understanding: an AI therapist that always agrees with a user may elicit positive short-term feedback, but it will not provide an effective form of therapy. Human feedback does not solve the problem in the pantry example either. A user may rate the Doritos restock positively in the moment, but the failure would only be revealed by later regret, which single-turn feedback does not capture.

### 3.3 Simulating human evaluations requires agents that already understand us

A third approach for evaluating model understanding is to simulate human participants *in silico* (Dubois et al., 2023; Horton, 2023; Binz et al., 2024; Argyle et al., 2023). Instead of recruiting human participants to interact with AI systems, the participants would themselves be simulated by other AI agents (such as multimodal LLMs). Evaluating AI understanding via simulated participants would marry the benefits of the previous two approaches: the scalability of automated metrics with the more realistic setting of human-like users.

This general approach has been the subject of recent excitement because of its potential to significantly scale up human evaluations (Anthis et al., 2025). However, while simulated agents might offer promise for some kinds of evaluations, there are significant challenges for using them to evaluate *understanding*. For one, the simulated agents must overcome the anthropomorphic bias and pluralistic alignment challenges faced in human-in-the-loop evaluation.

| Field | Core contribution | Open challenges |
|---|---|---|
| Theory of mind in AI | Benchmarks and models for attributing beliefs, desires, and knowledge to agents (Rabinowitz et al., 2018; Sap et al., 2022; Strachan et al., 2024) | Generalizing beyond vignettes about generic agents (Ullman, 2023; Shapira et al., 2024); interactive inference about a specific person over time |
| Plan and intent recognition | Recovering goals and plans from observed action traces (Kautz & Allen, 1986; Charniak & Goldman, 1993; Ramírez & Geffner, 2009) | Latent states beyond goals (emotions, misconceptions); noisy, long-horizon everyday behavior |
| User modeling and adaptive interfaces | Persistent models of a person's knowledge and preferences (Kobsa, 2001; Fischer, 2001; Jameson, 2008; Gajos et al., 2010) | Models that transfer across applications; inferring users' latent states from interaction logs |
| Context-aware computing | Sensing and responding to a user's situation (Abowd et al., 1999; Dey, 2001) | Individual-level understanding that persists across interactions and contexts |
| Preference learning and personalized alignment | Learning objectives from human feedback, increasingly at the level of individuals and groups (Christiano et al., 2017; Poddar et al., 2024; Jang et al., 2023; Sorensen et al., 2024; Kirk et al., 2024a) | Distinguishing stated, revealed, and reflective preferences; the inversion problem (Kleinberg et al., 2024) |
| Affective computing | Recognizing and responding to emotion from multimodal signals (Picard, 1997; D'Mello & Kory, 2015) | Interpreting what a recognized emotion means for a specific person in context |
| Computational cognitive science | Formal, mechanistic models of belief, desire, planning, and teaching (Tenenbaum et al., 2011; Baker et al., 2009; Jara-Ettinger et al., 2020; Griffiths et al., 2024) | Scaling structured models to open-ended interaction and integrating them with large-scale AI systems |

Table 2: Fields working toward parts of the grand challenge. Work in each field often involves observing a behavioral signal (actions, language, physiology) to infer a latent construct (goals, knowledge, affect) for some application (robots, interfaces, tutors). Building algorithms that understand us requires integrating across these fields and components.

Further, for AI-based simulations to effectively evaluate understanding, they must already have an understanding of humans. For example, they must know how we respond to certain stimuli and how those stimuli respond to internal states. There have been attempts to personalize the judgments of simulated humans (Dong et al., 2024), but creating high-fidelity simulations of individuals faces a fundamental data scarcity problem to accurately model personal idiosyncrasies (Park et al., 2024). Near-perfect simulation is not sufficient, as small errors in predictions of human behavior can result in large errors in measurement (Ludwig et al., 2025).

## 4 Progress in Improving Understanding

Characterizing evaluation protocols that assess understanding requires substantive, ongoing work, as we laid out above. Good evaluation paradigms are a means to the broader grand challenge: to be able to *engineer* AI systems that understand us. This quest too raises hefty challenges. But it is not impossible. There have already been glimmers of progress across different disciplines, as we lay out below, inspiring the way for more such work and innovation. Table 2 summarizes the fields contributing to this progress and the open challenges that remain for each.

**Collecting new kinds of data.** Section 3 demonstrated the drawbacks of using passively-collected behavioral data to evaluate understanding. However, efforts are under way to collect new kinds of data that offer higher-fidelity notions of understanding. For example, Park et al. (2024) collect two-hour interviews with 1,000 participants about their lives, values, and attitudes. They use this data to construct generative agents that are capable of replicating an individual's attitudes nearly as well as the individuals did two weeks later.

Researchers have also been collecting data that captures preferences and beliefs across different groups of people. For example, the PRISM Alignment dataset (Kirk et al., 2024b) links the sociodemographic profiles and stated preferences of 1,500 participants from 75 countries to their feedback on live LLM conversations—enabling culturally nuanced alignment objectives—while SubPOP (Suh et al., 2025) captures public opinion across 70K demographically representative U.S. subpopulations. Training models on this richer data can improve their understanding; for example, LLMs that are fine-tuned on SubPOP have 46% improvement in their ability to represent the opinions of diverse populations (Suh et al., 2025). Even basic categorical concepts such as color vary across cultures and languages, and recent similarity-judgment studies show that LLMs reflect this language-dependent variation (Marjieh et al., 2022; 2024; 2025; Sucholutsky et al., 2023b; Niedermann et al., 2024); accordingly, researchers are calling for richer datasets that capture both individual cognition and population-level diversity (Collins et al., 2023; Sucholutsky et al., 2023a; Ying et al., 2025).

These data need not be limited to text. Much of what people convey is nonverbal; visual and acoustic signals like facial expressions, tone of voice, and gaze carry information about latent states that words leave out. For example, a meta-analysis of affect detection systems found that multimodal systems were more accurate than their best unimodal counterparts in 85% of the systems surveyed (D'Mello & Kory, 2015). As multimodal AI systems mature (Baltrušaitis et al., 2019), collecting datasets that pair these signals with people's underlying states, a longstanding goal of affective computing (Picard, 1997), offers another path toward models that understand us. Data collection can be difficult to incentivize, as algorithmic or modeling contributions are often prioritized by AI researchers (Gero et al., 2023), but it is clear that there is much more fruitful work to be done in this domain.

**Personalizing understanding.** The same action can come from different latent states in different people. For example, one person might decline a social invitation because of anxiety while for somebody else it might be because of a scheduling conflict. However, most alignment pipelines average preference data across annotators, producing models of a mean person (who might not even exist) that do not capture individual and cultural variation (Siththaranjan et al., 2023; Casper et al., 2023; Kirk et al., 2024b). Recent work has started to mitigate this by treating personalization as an objective in its own right. Variational preference learning infers a distribution over individual reward functions (Poddar et al., 2024), and parameter-merging methods learn separate policies for distinct preferences and mix them to match an individual (Jang et al., 2023). Benchmarks like LaMP test whether model outputs actually adapt to a specific user (Salemi et al., 2024). Researchers working on pluralistic alignment have made progress on the broader question of what it means to serve many people whose values differ (Sorensen et al., 2024; Kirk et al., 2024a). One useful framing is hierarchical, where population data provides a prior over how people vary, and understanding an individual can be thought of as making inferences under that prior from limited interaction. Any one person provides orders of magnitude less data than a pretraining corpus, so personalization is largely a problem of sample efficiency. However, we reiterate that a system that has learned to match how a person talks has not necessarily modeled why they act as they do. Personalization also heightens risks since the same person-specific models that make assistance useful can raise the privacy and manipulation concerns that we discuss in Section 5.

**Incorporating insights from the behavioral sciences.** Addressing challenges like the inversion problem from Section 3 requires incorporating high-fidelity models of human behavior. For example, to understand whether a user who is about to eat Doritos actually wants them, an algorithm must be able to translate the user's behavior into an internal state: is the user's behavior consistent with previous behavior from when they make deliberative, well-thought-out choices? Or is it more consistent with their behavior when they make rushed choices that they later regret?

Fortunately, there's a field full of insights about human behavior: the behavioral sciences. Researchers have already shown promising results by incorporating these insights into AI models. Agan et al. (2023) address the inversion problem above by inferring and disentangling System I (fast, automatic) behaviors from System II (slow, deliberative) judgments. In robotics, Sripathy et al. (2022) demonstrate that a cognitively-inspired machine learning model can capture affective states in robot motion. Insights from the behavioral sciences have also been used to encourage LLMs to produce text that is better aligned with human preferences. A common strategy is to perform reinforcement learning from human feedback (RLHF) (Christiano et al.,

2017) using the Bradley-Terry model (Bradley & Terry, 1952), a model of human preferences. Ethayarajh et al. (2024) demonstrate that this model is incomplete, and draw on prospect theory (Kahneman & Tversky, 2013) from behavioral economics to improve the alignment of LLMs.

The behavioral and cognitive sciences also offer insights into incorporating intellectual understanding into AI systems. In educational settings, a machine could be built from the ground up to explicitly model the set of students' misconceptions and simulate what "worlds" may have led to someone's (mis)understanding in the first place, baking in insights from cognitive science about what makes for good explanations (Chandra et al., 2024; Lombrozo, 2006; Miller, 2019). For the student struggling with trigonometry from the introduction, such a system would not only predict which problems they get wrong, but also infer the underlying misconception, such as a belief that cosine is always positive, and construct a lesson that addresses it (Rafferty et al., 2020; Ross & Andreas, 2024). More broadly, decades of computational cognitive science work have made advances formally modeling a suite of human cognitive abilities from how we plan (Ho et al., 2022; van Opheusden et al., 2023) to how we reason about each others' beliefs and desires (Baker et al., 2009; 2017; Jara-Ettinger et al., 2020) to how human teachers infer students' misconceptions (Rafferty et al., 2020), all of which can inform – in computational terms – the engineering of human-AI thought partnerships (Collins et al., 2024d) or other kinds of machine "cognitive prostheses" (Lieder et al., 2019; Callaway et al., 2023). For example, one engineering approach is to build hybrid neuro-symbolic architectures that enable new kinds of interpretable and flexible understanding of human behavior (e.g. by capturing how we understand each other) (Zhi-Xuan et al., 2020; 2024; Ying et al., 2023).

One kind of intellectual understanding is especially important for the performance of AI systems: in order for AI to understand people, it must also understand how people understand AI (Steyvers & Kumar, 2023; Gweon et al., 2023; Bansal et al., 2019). Plenty of work in the behavioral and cognitive sciences has studied this reverse question, developing tools to measure and improve how people understand algorithms (Kelly et al., 2023). Researchers have begun incorporating these insights to improve AI systems. For example, Vafa et al. (2024b) build a machine learning model to predict how people would deploy LLMs based on short interactions, thereby enabling evaluation of LLMs based on how people would likely deploy them.

**Building AI systems with the right inductive biases for human behavior.** Applications of AI methods to behavioral data typically use off-the-shelf machine learning models such as multilayer perceptrons with generic initialization. This approach treats behavioral data as just another kind of data, deploying the same models that would be used if we were trying to make predictions in any other scientific domain. As a consequence, significant amounts of data are typically needed to train these models to make good predictions.

In other settings, the amount of data needed to train AI systems has been reduced by making use of models that have inductive biases that are compatible with data in that domain. For example, early progress in image classification benefited from the use of convolutional neural networks (LeCun & Bengio, 1995; LeCun et al., 1995), which build in the expectation that effective features for classification would be invariant to spatial translation within images. Likewise, a variety of technologies for processing text have made use of neural network architectures that build in an expectation that more recent information is more likely to be useful for predicting the next word (Elman, 1990; Hochreiter & Schmidhuber, 1997). Are there analogous inductive biases that can be drawn upon for creating AI systems that need less data to make good predictions about human behavior?

Psychological theories may provide an effective source for such inductive biases. In the domain of decision-making, a series of papers have explored how features derived from psychological models can be used to make better predictions about human decisions (Plonsky et al., 2017). Pretraining neural networks on synthetic data generated from psychological theories also proves effective in reducing the amount of human data required (Bourgin et al., 2019). Theories like expected utility maximization or prospect theory can also be used to constrain the functional form of neural network architectures, resulting in differentiable decision theories that can be optimized using tools from machine learning (Peterson et al., 2021). Similar approaches might be used to translate psychological theories into effective inductive biases in other domains of human behavior.

# 5 Benefits and Risks of Improved Understanding

We stand to gain many potential benefits from progress towards this grand challenge. The most direct benefit is improved AI tools: AI assistants could write emails and make purchases on our behalf; AI tutors could craft lesson plans tailored specifically to our misunderstandings; AI therapists could help us navigate decisions whenever we need them. However, the potential benefits go far beyond better tools. We consider some additional benefits here, but emphasize that progress towards this quest of algorithms that understand is not necessarily intended to have a single "solution" nor provide an instant salve toward many other problems. Rather, it is likely a continually evolving challenge as a core component – people – fundamentally change and continue to evolve, too.

**Shared understanding.** While the grand challenge we emphasize here involves building algorithms that understand people, a long literature in machine learning has studied the reverse question: building interpretability methods to improve people's understanding of models (Zeiler & Fergus, 2014; Rahwan et al., 2019; McCoy et al., 2024; He et al., 2024; Ku et al., 2025). If people understand models and models understand people, it can support a richer *shared, or mutual, understanding* (Figure 2). Shared understanding would enable a new, reliable mode of communication between people and machines (Bobu et al., 2024). These benefits may accrue if models can develop a personalized understanding of their users, creating higher efficiency or utility, and permitting technologies that work for the many, not the few (Kirk et al., 2024a). This mutual understanding — where we understand machines, the machines understand us, and together we understand the world and task at hand — can lay the foundations of a new class of rich human-AI thought partnerships (Collins et al., 2024d; 2026).

**Advancing behavioral science with AI.** If algorithms understand people, this provides an opportunity to advance the behavioral and cognitive sciences. For example, LLMs are increasingly being leveraged in computational social science (Ziems et al., 2024) and psychology (Rathje et al., 2024) for tasks that previously required domain expertise; Luo et al. (2025) even suggest that LLMs may be better than neuroscientists at predicting the outcomes of neuroscience experiments. This invites an important question: what if AI systems understand people better than our theories currently do?

There has already been progress in incorporating AI to improve theories about people (Bobu et al., 2020). For example, Peterson et al. (2021) and Mullainathan & Rambachan (2024) train machine learning models on lottery choice data: given the choice of lotteries with different tradeoffs, these approaches use machine learning procedures to predict which lottery a human participant would choose. These models are then used to refine behavioral theories about how people make choices. For example, Mullainathan & Rambachan (2024) discover new anomalies for expected utility theory from neural networks trained on this lottery choice data. The use of more human-aligned AI systems can also inform our theories of cognition (as outlined in Section 4). Moreover, AI systems that understand us open up a range of further studies into *networks* of interactions (Collins et al., 2025; Shiiku et al., 2025; Chang et al., 2025b; Sucholutsky et al., 2025). This is important not only to deploy AI systems into realistic diverse settings, but also to inform our understanding of human group behavior and cultural transmission (Brinkmann et al., 2023; Shin et al., 2023).

The benefits do not end at improved understanding of ourselves. If we use theories from the behavioral sciences to improve AI systems and then use those systems to improve theories from the behavioral sciences, this creates a *feedback loop*, where each keeps improving (see Figure 2). Such reciprocal enhancement could accelerate progress between both progress towards the grand challenge of AI that understands us, and the fields contributing to it – leading not only to more effective AI tools but also to deeper and more nuanced theories of human behavior.

**Towards better policy and decision making.** If AI systems understand us, we can consider using such models to inform counterfactual policy making, e.g., predicting how people may act in response to particular interventions. Determining the impact of interventions on society is a classically wicked problem (Rittel & Webber, 1973), and has been around nearly as long as this kind of decision making has been around (Merton, 1936). AI systems that understand us could be used to help design better protocols around human-AI interaction (Koster et al., 2022) or other "infrastructure" around human-AI interaction (e.g., whether to impose a nudge (Callaway et al., 2023) or friction (Collins et al., 2024a), from knowing that we need some "push" to either use a tool more or less). While behavioral science is ultimately a pursuit of knowledge, the

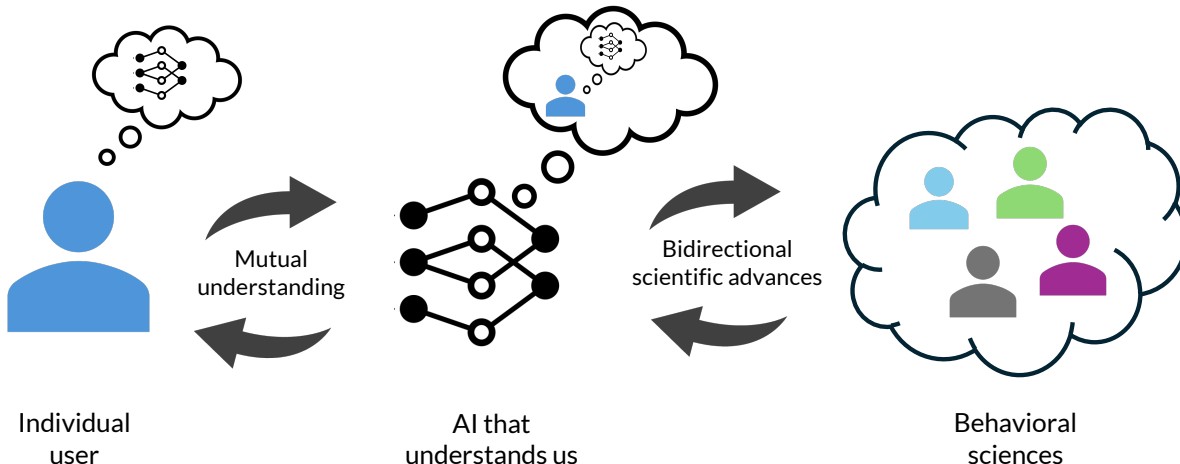

Figure 2: **Feedback loops that may arise from progress towards building algorithms that understand us.** Any one person interacting with AI systems may learn about the model they are interacting with, while the model develops its understanding of the user (and the user's understanding of it). This cycle can give rise to produce mutual understanding. In tandem, advances in building algorithms that understand us can inform the behavioral sciences, which in turn can feed back to the design of better AI that understands us.

interplay between behavioral sciences and AI has the potential to aid policy and decision making, though is just one piece of the broader complexity of engaging with potential societal interventions, as we discuss next.

**Risks.** While there are many benefits that can arise from AI systems that understand us, those pursuing this grand challenge of building algorithms that understand us must also engage with and take steps to alleviate the risks. One risk is that there may be a limit to how much people want AI systems to understand them. Opening up AI systems to understand us requires addressing trade-offs in privacy (Kirk et al., 2024a). Are there cases then where we do *not* want an AI system to understand us? How does our ability to understand models inform which models we choose to use? More broadly, if one can model people, then such models can be leveraged to optimize a social outcome. If the goals of Behavioral AI are achieved, systems that understand us and even our theory of mind, we anticipate such developments could also lead to changes in personalized advertising, political persuasion, emotional manipulation, fraud, deception, and information extraction, among other societally deleterious use cases (Kirk et al., 2024a; Hackenburg & Margetts, 2024; Matz et al., 2024; Hagendorff, 2024; Meguellati et al., 2024; Schoenegger et al., 2025).

Rather than being incidental, these risks stem from the same capabilities researchers across different fields may pursue when aiming to build systems that understand people. A system that can infer what a user feels can also infer when they are most persuadable, enabling manipulation and persuasive messaging tailored to each individual (Matz et al., 2024; Hackenburg & Margetts, 2024). Systems that model our emotions can result in emotional dependence, with users forming attachments to AI companions that displace human relationships (Laestadius et al., 2024; Fang et al., 2025). Being able to infer someone's latent states also changes the privacy risks. Exposed information may no longer be limited to what a person has chosen to share, as it can also include things they never chose to disclose. Nor do these harms require a malicious actor. A model whose objective diverges from its users' interests, as when systems are optimized for engagement or approval, can turn its understanding against the very people it models; models optimized on user feedback have already been shown to learn targeted manipulation and deception (Williams et al., 2024; Hagendorff, 2024). Even the pantry example from Section 3 illustrates these risks: a system that knows when its user's resolve is weakest could use that knowledge to support the user's goal or to exploit it.

These risks suggest concrete safeguards and design principles that should develop alongside any efforts towards this grand challenge. Users should be able to see, correct, and delete what a system infers about them, and systems should be transparent about which latent states they model. This connects to the growing literature on the editability of AI systems, e.g. methods for editing knowledge in LLMs (Meng et al., 2022; Yao et al., 2023). Data collection can draw on privacy-preserving methods like federated learning (McMahan et al., 2017) and differential privacy (Dwork & Roth, 2014), which let models learn about people in aggregate while limiting what is retained about any individual. When systems act on inferred states, they should be optimized for users' own goals rather than for engagement or persuasion (failures like sycophancy show how easy this line is to cross (Sharma et al., 2025)). In some settings, the right design choice is to not infer at all. Determining how these systems should be governed, including which inferences require consent and which should be off limits entirely, is as central a question for Behavioral AI as the technical ones.

As with any new technology, there are many possible uses: some good, some bad. However, as with any field that engages with people, we believe we need a broad tent of voices to engage with these challenges and shape the direction of the technology well. Coalescing researchers across distinct fields that are each working toward the grand challenge of building and assessing Behavioral AI can facilate more of these critical cross-disciplinary conversations.

## 6 Conclusion

Building algorithms that systems that can understand us is a grand challenge. Progress towards this challenge demands an interdisciplinary effort, combining insights from computer science and the behavioral sciences. However, assembling interdisciplinary teams is not enough; making progress requires researchers who are "bilingual," fluent in both understanding algorithms and behavioral science. To fully realize this vision, we must develop robust frameworks for evaluating understanding and then produce new tools that succeed at these evaluations, and help us realize where our current evaluations fall short. In doing so, efforts towards building and assessing Behavioral AI can in turn transform our relationship with AI, creating not just more effective tools, but also fostering deeper scientific insights into human behavior itself.

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
