# OpenReview forum: "Behavioral AI: Building Algorithms That Understand Us"
_TMLR — Under review for TMLR_

### Review · Reviewer_1k6T · 2026-05-13

**Summary Of Contributions:**

The paper presents a perspective on behavioral AI. It presents current issues of AI in understanding human preferences, emotions and intellect, challenges in evaluating such understanding and promising directions to overcome them. They end with discussing benefits and risks of such a AI understanding Human Behavior.

Strengths:
- The paper is clear and easy to follow.
- It clearly presents the issues with current AIs in understanding human behavior, and challenges in evaluating such understanding.
- The paper presents future paths to explore to build an AI with human behavior understanding.
- The paper presents the risks and benefits of such an AI.

Weaknesses:
- The paper does not discuss multimodal AI which is crucial for understanding Human behavior as visual and acoustic signals often provide more information about latent preferences.
- The paper has very short discussion on personalization, which is very important as behavior of people manifest in very personalized ways.
- The paper has a very short discussion on the risks. The paper should expand on how they maybe exploited by malicious parties influencing people or the models with such understanding become misaligned.
- The paper does not have a discussion on how easy it is for humans to understand the behavior of other humans (from different countries/cultures vs same country/culture) or how much time or number of interactions it takes humans to understand behavior of other humans. Similarly the benchmark evaluations do not discuss such a human baseline.

**Audience:**

Yes

**Audience Explanation:**

The paper presents an interesting perspective on advancing AI's understanding of human behavior. While the paper itself does not present a new methodology, dataset or benchmark for the same it discusses the issues with current AIs, evaluation challenges, possible advancement avenues and risks and benefits of such an AI. To the best of my knowledge, this a fresh perspective on understanding human behavior through AI.

**Claims And Evidence:**

Yes

**Claims Explanation:**

The authors provide citations for there claims.

**Requested Changes:**

- The paper should discuss multimodal AI in understanding Human Behavior.
- The paper should include more discussion on personalization.
- The paper should include more discussion on the risks.
- The paper should include a discussion on how easy it is for humans to understand the behavior of other humans (from different countries/cultures vs same country/culture) or how much time or number of interactions it takes humans to understand behavior of other humans and discuss them as human baselines in evaluation section.

---

> ### Author Response · Authors · 2026-07-17
> **Response to Reviewer 1k6T**
>
> Thank you for your positive review. We respond to each of your comments below.
>
> > _The paper does not discuss multimodal AI which is crucial for understanding Human behavior as visual and acoustic signals often provide more information about latent preferences._
>
> We agree, and we've added a discussion to the paper about how visual and acoustic signals carry information about latent states that text leaves out. Section 4 now discusses multimodal signals as a source of higher-fidelity behavioral data, connecting to a long line of work on affective computing and multimodal machine learning.
>
>
> > _The paper has very short discussion on personalization, which is very important as behavior of people manifest in very personalized ways._
>
> We agree and have now added a dedicated discussion of personalization to Section 4. We also connect personalization to the expanded privacy discussion in Section 5, since personalized models can raise privacy and manipulation concerns.
>
> > _The paper has a very short discussion on the risks. The paper should expand on how they maybe exploited by malicious parties influencing people or the models with such understanding become misaligned._
>
> We agree that the risks deserve a deeper treatment, and we've expanded the risks discussion in Section 5. It now directly discusses manipulation and persuasion tailored to inferred states, emotional dependence, privacy harms and exploitation by malicious actors, and misalignment, where systems optimized for engagement or approval turn their understanding against the people they model. It also lays out concrete safeguards and design principles, including user control and transparency over what a system infers, privacy-preserving data collection, limits on persuasive optimization, settings where systems should not infer at all, and the governance and consent questions these systems raise.
>
>
> > _The paper should include a discussion on how easy it is for humans to understand the behavior of other humans (from different countries/cultures vs same country/culture) or how much time or number of interactions it takes humans to understand behavior of other humans and discuss them as human baselines in evaluation section._
>
> This is a nice suggestion. Understanding other people is difficult even for humans; classic results in psychology show that people read emotions more accurately within their own culture than across cultures, and that friends infer each other's thoughts and feelings more accurately than strangers. We've added these findings to the evaluation section (Section 3.2), noting that human performance suggests one baseline for evaluating understanding, though not necessarily a ceiling, since machines may come to understand people in ways that differ from how we understand each other.

---

### Review · Reviewer_vRwg · 2026-05-27

**Summary Of Contributions:**

This (perspective) paper proposes "Behavioral AI" as a new interdisciplinary subfield focused on developing AI systems that understand humans across preferential, intellectual and emotional dimensions. The paper argues that, while AI research has historically emphasized helping humans understand models, the next generation of AI systems will require the reverse. The manuscript surveys related efforts across NLP, robotics, alignment, HCI and cognitive science, discusses challenges in evaluating "understanding," and outlines possible research directions involving richer data collection, behavioral-science-informed modeling, and human-centered inductive biases.

The topic is timely and important, and the paper raises several (possibly too many) worthwhile questions about human-AI interaction, personalization and latent-state inference. However, the paper ultimately feels much shallower than its framing initially suggests. The central concept of "Behavioral AI" is asserted more than established. The manuscript does not clearly explain why this framing is distinct from existing areas such as human-centered AI, user modeling, adaptive AI, affective computing, theory-of-mind modeling, plan recognition, or preference learning. More importantly, the paper undermines its own terminology by explicitly arguing that "many important human states are **not** captured by behavior", while still anchoring the proposed field around the term "behavioral." This tension is never satisfactorily resolved.

The paper generally suffers from excessive breadth, conceptual repetition, and unusually weak editorial polish (a *genuine* one rather than a quick (AI) polishing pass) for a perspective article. The writing often substitutes expansive rhetoric for analytical precision, and the bibliography contains multiple formatting anomalies and duplicated entries. There is additionally a likely **double-blind violation** in the conclusion.

**Additional Comments:**

I appreciate the ambition of the paper and agree that the broad problem is important. My negative assessment is not based on the absence of experiments or technical contributions, and I have tried to carefully evaluate it as a perspective/position paper. My naub concern is that this manuscript currently does not yet provide the level of conceptual precision, scholarly positioning, or argumentative discipline expected of a strong perspective article.

The strongest version of this paper would not simply announce a new field. It would explain why existing fields fail to provide the needed conceptual structure, identify the precise object of study, define the main technical and ethical bottlenecks, and give researchers a clear agenda. At present, the paper is closer to a broad essay collecting related themes than to a rigorous perspective that reorients the literature.

**Audience:**

Yes

**Audience Explanation:**

The underlying topic is indeed (very) relevant to the TMLR audience. Various researchers across AI-focused, human-focused as well as human+AI-focused domains are actively grappling with how AI systems should model and respond to human preferences, goals, emotions, and reasoning processes. The paper does raise useful questions about the limits of behavioral prediction and the difficulty of evaluating (latent) understanding.

However, interest in the *topic* should not be conflated with the strength of the present manuscript. In its current form, the paper reads more like a broad collection of observations and references than a sharply argued (perspective) article. A stronger version would provide a more precise conceptual framework, clearer differentiation from adjacent literatures, and a substantially more actionable research agenda.

**Broader Impact Concerns:**

The paper discusses several risks, but the broader impact discussion should be more substantial given the topic. Systems designed to infer latent preferences, emotions, vulnerabilities, and intentions could enable genuinely useful applications in tutoring, accessibility, and assistive technologies. The same capabilities could also support manipulation, emotional dependency, hyper-personalized persuasion, political targeting, surveillance, and exploitation of psychologically vulnerable users.

These risks are not peripheral, but structurally tied to the paper's central vision of systems that understand humans. The manuscript should therefore discuss governance, consent, privacy, user control, and limits on persuasive optimization in a much more explicit and systematic way.

**Claims And Evidence:**

No

**Claims Explanation:**

The paper does not need to provide new experiments, which is fine since it is explicitly framed as a perspective piece. However, a perspective paper still needs to support its claims through conceptual clarity, accurate synthesis of prior work, careful positioning, and a compelling argument that the proposed framing adds value beyond existing literatures. I do not think the current manuscript reaches that standard.

First, the core claim about (need for) a new subfield called Behavioral AI is not adequately justified. The paper cites many relevant examples of systems that infer preferences, model user behavior, personalize interactions, or incorporate ideas from behavioral science. However, it does not explain why these examples cohere specifically under the term "Behavioral" AI, nor does it delineate the proposed field from already established areas such as user modeling, affective computing, human-centered AI, computational cognitive science, plan recognition, theory of mind in AI, preference learning, recommender systems, personalized alignment, human-robot interaction, and adaptive interfaces. The paper therefore shows that there is broad activity around AI systems and humans, but it does not convincingly show that the proposed conceptual category is necessary, non-redundant, or scientifically clarifying.

Second, the term "understanding" is doing too much work. The paper sometimes uses it behaviorally, as in systems that act *as if* they understand users; sometimes cognitively, as in inferring latent beliefs, preferences, emotions, or goals; sometimes instrumentally, as in improving downstream interaction quality; and sometimes scientifically, as in producing better theories of human behavior. These are related but not equivalent. Without a clearer account of what kind of understanding is being claimed, the argument becomes difficult to evaluate. The paper gestures at some distinctions, especially in its discussion of the inversion problem, but does not build them into a coherent framework.

Third, much of the evidence is illustrative rather than analytic. The examples involving AI tutors, writing assistants, emotional companions, and AI agents are plausible and relevant, but they often function as motivating anecdotes rather than as evidence for a well-specified research agenda. Similarly, the literature survey contains many appropriate citations, but it is organized more as a catalog of related developments than as a synthesis that identifies gaps, tensions, definitions, or testable research questions.

Fourth, some claims are too sweeping relative to the support provided. The manuscript repeatedly suggests that Behavioral AI could improve AI tools, advance behavioral science, support better policy, and enable richer human-AI partnerships. These are possible benefits, but the current draft does not sufficiently specify the conditions under which they would hold, what the main technical bottlenecks are, or how one would know the field is making progress. The paper would be stronger if it were more explicit about tradeoffs, impossibility results or lower bounds where relevant, and cases where more "understanding" may be undesirable.

Fifth, the scholarly execution weakens confidence. The reference list and citation formatting appear insufficiently proofread. Examples include malformed author lists where "et al." appears inside the list of named authors, inconsistent citation style, and duplicated references. These issues are not merely cosmetic in a perspective paper, because the paper's main contribution is its synthesis of prior work. If the synthesis is the contribution, the bibliography and positioning need to be especially careful.

**Requested Changes:**

## Critical changes required for acceptance

### **Clarify and justify the central terminology, and strengthen positioning relative to existing literature.**
The paper needs to explain why "Behavioral" AI is the correct framing given adjacent established fields (like human-centered AI, personalized AI, preference learning, affective computing) as well as emerging, existing constructs like behavior-aware AI (https://openscholarship.wustl.edu/eng_etds/1080/, https://www.ijcai.org/proceedings/2024/344) or context-aware AI (https://arxiv.org/abs/2512.24362). It also needs to align its own criticism of behavior at times being an insufficient proxy for latent human states. and its choice of "behavioral" AI terminolgoy that currently feels internally inconsistent. The authors should either defend this framing rigorously or reconsider it entirely.

A case can also be made for overclaiming. The manuscript often presents Behavioral AI as a broad solution to (too) many problems in AI systems, behavioral science, and policy. The paper should more carefully state what improved human modeling can and cannot solve. It should also identify cases where systems should *not* infer more about users, where user understanding may be harmful, and where optimizing for inferred latent states could produce paternalistic or manipulative systems.

The manuscript should engage much more directly with existing work on theory of mind (in AI), plan recognition (in robotics), user modeling, adaptive interfaces, personalized alignment, and computational cognitive science. At present, the paper often presents ideas as newer or less explored than they actually are.

### **Sharpen the conceptual framework. Reducing breadth and rhetorical repetition could be helpful.**
The paper needs a clearer account of what "understanding" means *operationally*. The manuscript currently moves fluidly between prediction, latent-state inference, personalization, successful assistance, and scientific explanation without distinguishing them carefully enough, which is critical in absence of other technical contributions.

The paper attempts to simultaneously discuss tutoring, writing assistants, emotional companions, social simulation, behavioral science, policy, interpretability, and alignment. The result is diffuse. Broad vision/framing is not inherently problematic--and could even be a major plus point in some perspective/position paper--but this paper falls short of adequately handling the breadth. The authors should either narrow the manuscript--for example, to "behavioral-science-informed design of human-centered AI"---or impose a much stronger organizing structure.

### **Improve scholarly and editorial quality.**
The bibliography requires a thorough audit. There are duplicated references, malformed author lists (e.g., using "et al" in middle of author names like Shane et al Legg, Thomas F et al Müller, Esin et al Durmus), inconsistent citation formatting, and several signs of insufficient proofreading--or even possibly a *problematically* AI-assisted (hallucinated) references/paper preparation that has recently led to desk rejections at top research venue. The prose itself also requires tightening, and the manuscript has telltale signs of AI abuse (or borderline "AI slop") relies heavily on stylistic devices (especially over 30 colons in main text, often used suboptimally if not incorrectly, and em dashes). **This coupled with the statement referencing the author's team hosting a workshop on this at NeurIPS 2024 that can compromise anonymity could be ground for desk rejection**.

## Additional changes that would strengthen the work

1. A concise comparison table situating Behavioral AI relative to adjacent fields would greatly improve clarity. The manuscript would be stronger if reframed more modestly around the idea that behavioral and cognitive sciences should inform the design and evaluation of human-centered AI systems, rather than positioning ``Behavioral AI'' as an entirely new field.

2. The paper would benefit from one or two deeper case studies rather than many shallow examples.

3. The risks section should more directly, and deeply, discuss manipulation, persuasion, emotional dependence, and privacy harms arising from systems that infer latent human states.

4. Make the research agenda actionable. At present, the paper identifies broad desiderata but gives insufficient guidance for researchers entering the area. The paper should articulate concrete open problems, for example:
   - What are benchmarkable tasks for preferential, intellectual, and emotional understanding?
   - What data would be required to distinguish stated preferences from latent preferences?
   - How should longitudinal user modeling be evaluated?
   - What privacy or consent constraints should govern models that infer latent mental states?
   - What failure modes distinguish poor behavioral prediction from poor latent-state inference?
   - What would constitute progress beyond standard personalization?

---

> ### Author Response · Authors · 2026-07-17
> **Response to Reviewer vRwg (1/2)**
>
> Thank you for your detailed review. We've made substantial changes in response, and we respond to each point below.
>
> > _The paper needs to explain why "Behavioral" AI is the correct framing given adjacent established fields (like human-centered AI, personalized AI, preference learning, affective computing) as well as emerging, existing constructs like behavior-aware AI or context-aware AI... The authors should either defend this framing rigorously or reconsider it entirely... The manuscript often presents Behavioral AI as a broad solution to (too) many problems in AI systems, behavioral science, and policy. The paper should more carefully state what improved human modeling can and cannot solve._
>
> Thank you for the thoughtful comments. They have triggered an active discussion among some of our author team and led us to realize that, indeed, the paper overclaimed that Behavioral AI is a new field. Instead, we have reframed our work as clarifying that the quest to build algorithms that understand us is a “grand challenge.” Many subfields are working toward parts of this grand challenge, although often in isolation; our goal is to coalesce these efforts. We have done a substantive rewrite of the framing of the paper around this grand challenge (one framed then as not necessarily having a “definitive solution”) and needing a big tent.  The introduction now engages the adjacent fields directly, and Table 2 situates each field's contributions and open challenges relative to the grand challenge. We have also tempered claims about what improved human modeling can solve, and the expanded Section 5 now identifies cases where systems should not infer more about users and where optimizing for inferred states becomes manipulative.
>
> > _It also needs to align its own criticism of behavior at times being an insufficient proxy for latent human states. and its choice of "behavioral" AI terminolgoy that currently feels internally inconsistent._
>
> The potential tension between behavior and latent states is a good point. We see this tension as the motivation for Behavioral AI rather than an inconsistency in it. Behavior is the primary signal current AI systems observe about people, and so a central challenge is recovering the latent states beneath it. The name mirrors behavioral economics, which also starts from observed behavior and derives its insights from the ways behavior departs from underlying preferences [1]. Thank you for pointing this out, and we've added more about this to Section 3.1, which we feel strengthens the thrust of the paper.
>
> > _The manuscript should engage much more directly with existing work on theory of mind (in AI), plan recognition (in robotics), user modeling, adaptive interfaces, personalized alignment, and computational cognitive science._
>
> We agree the paper needed to engage these literatures more directly. We have added a discussion of adjacent fields to Section 1 that engages machine theory of mind, plan and intent recognition, user modeling and adaptive interfaces, context-aware computing, personalized and pluralistic alignment, affective computing, and computational cognitive science. We make it clear that these literatures present longstanding work toward a shared grand challenge, and we added Table 2 to make each field's contribution and the remaining challenges more clear.
>
>
> > _The paper needs a clearer account of what "understanding" means operationally. The manuscript currently moves fluidly between prediction, latent-state inference, personalization, successful assistance, and scientific explanation without distinguishing them carefully enough...the term "understanding" is doing too much work._
>
> Thank you for raising these issues (shared with zoWw). To clarify how we use the term: our claim is functional. As the introduction states, systems need not understand the way humans do, but they should behave as if they understand, and the beginning of Section 3 now defines this explicitly: a system understands a person to the extent that it recovers the latent states driving their behavior (preferences, knowledge, and emotions) well enough to support their goals. Prediction, personalization, and successful assistance are then measurable proxies for this target, and Section 3 examines when these proxies do and do not track it. Producing better theories of human behavior is a potential byproduct we discuss in Section 5 rather than part of the definition.
>
> We have added a new table (Table 1) which aims to concretize the dimensions of understanding further. We have also added additional caveats in the section that these dimensions can interact (and are not necessarily the _only_ dimensions of understanding). We have also adjusted the beginning of Section 3 to clarify that understanding is not easy to operationalize, thereby motivating work towards this grand challenge.
>
> [1] Thaler, R H. "Behavioral economics: Past, present, and future". American Economic Review, 2016.

---

> > ### Author Response · Authors · 2026-07-17
> > **Response to Reviewer vRwg (2/2)**
> >
> > > _The bibliography requires a thorough audit. There are duplicated references, malformed author lists (e.g., using "et al" in middle of author names like Shane et al Legg, Thomas F et al Müller, Esin et al Durmus), inconsistent citation formatting, and several signs of insufficient proofreading--or even possibly a problematically AI-assisted (hallucinated) references/paper preparation..._
> >
> > Thank you for catching these. We've gone through the bibliography and fixed these issues. The malformed author lists were not LLM artifacts (every reference is a real paper) but rather formatting errors from our paper library software. We've fixed all affected entries, removed duplicated entries, and made the formatting consistent across the bibliography (e.g. capitalization of acronyms in titles and venue names).
> >
> >
> > > _The prose itself also requires tightening, and the manuscript has telltale signs of AI abuse (or borderline "AI slop") relies heavily on stylistic devices (especially over 30 colons in main text, often used suboptimally if not incorrectly, and em dashes)._
> >
> > We did not rely on AI systems or LLMs to write this paper. While we used them to proofread (in keeping with TMLR policy), the writing is ours. There are a lot of colons and em dashes because we use them liberally in our writing. That said, we agree the prose benefits from tightening, and we've cleaned this up in the revision.
> >
> >
> > > _The statement referencing the author's team hosting a workshop on this at NeurIPS 2024 that can compromise anonymity could be ground for desk rejection._
> >
> > You're right, and we apologize for this oversight. We've removed the sentence from the conclusion. Thank you for flagging it.
> >
> >
> > > _1. A concise comparison table situating Behavioral AI relative to adjacent fields would greatly improve clarity..._
> >
> > Great point, we've now added this table (Table 2).
> >
> >
> > > _2. The paper would benefit from one or two deeper case studies rather than many shallow examples._
> >
> > We agree that having case studies running through the whole paper would be helpful for readers. We have added more discussion of our main two illustrative case studies of the smart pantry of Kleinberg et al. (2024) and the trigonometry student from the introduction, and now refer back to them throughout the whole paper. We're happy to expand either example further if the reviewer prefers.
> >
> > > _3. The risks section should more directly, and deeply, discuss manipulation, persuasion, emotional dependence, and privacy harms arising from systems that infer latent human states... The manuscript should therefore discuss governance, consent, privacy, user control, and limits on persuasive optimization in a much more explicit and systematic way... It should also identify cases where systems should not infer more about users, where user understanding may be harmful, and where optimizing for inferred latent states could produce paternalistic or manipulative systems._
> >
> > We agree that the risks deserve a deeper treatment, and we've expanded the risks discussion in Section 5. It now directly discusses manipulation and persuasion tailored to inferred states, emotional dependence, privacy harms and exploitation by malicious actors, and misalignment, where systems optimized for engagement or approval turn their understanding against the people they model. It also lays out concrete safeguards and design principles, including user control and transparency over what a system infers, privacy-preserving data collection, limits on persuasive optimization, settings where systems should not infer at all, and the governance and consent questions these systems raise.
> >
> >
> > > _4. Make the research agenda actionable. At present, the paper identifies broad desiderata but gives insufficient guidance for researchers entering the area. The paper should articulate concrete open problems._
> >
> > We agree the agenda should have been more concrete, and we have revised accordingly. We have added Table 2 which lists open challenges for each contributing field and Table 1 which maps each dimension of understanding to representative tasks, failure modes, and possible evaluation approaches (benchmarkable tasks). Section 3.1 develops the gap between predicting behavior and inferring latent states through the pantry example and Section 3.2 shows why single-turn feedback misses the relevant signal and discusses human performance as a baseline. The new personalization discussion in Section 4 frames personalization as sample-efficient inference under a population prior, and Section 5 discusses which inferences should require consent or be off limits entirely (privacy/consent constraints).

---

### Review · Reviewer_zoWw · 2026-06-28

**Summary Of Contributions:**

This submission presents Behavior AI as a perspective and research field for building AI system that can better understanding humans. An interesting point is that it shifts attention from conventional focuses on foundation model improvement in task completion to reverse question of whether AI systems can understand human preferences, emotion situations and hidden beliefs. It structurally discusses current deficiencies in AI's understanding of people, challenges in measuring such understanding, recent progresses from data collection and modeling informed by behavior-science, and the potential benefits and risks of developing AI systems that better understanding humans.


Strengths:

It offers a timely and valuable perspective on mutual Ai understanding humans and reverse human understanding of AI. The paper motivates this perspective with many intuitive examples, such personalized tutoring, implicit preference inference and emotional support, which make the argument concrete and accessible. It is also clearly written and concise, covering a broad topic of recent research works without becoming unnecessarily long. Another strength is its multidisciplinary framing, connecting AI with behavior science and preference learning, which contributions can be meaningful to various fields.

Weakness:

The main weakness is that the aspect of Behavior AI as a distinct research field is not very novel. Many of the topics discussed in the paper already exist in related areas such as human-AI interaction and behavior science, so the paper should more clearly explain what Behavioral AI adds beyond serving as a grand term. In addition, while the paper presents a compelling high-level agenda, it remains abstract and would be stronger with more concrete guidance on evaluation protocols, technical formulations or actionable benchmarks for measuring AI's understanding of humans.

**Audience:**

Yes

**Audience Explanation:**

In my opinion, at least some individuals in the audience would likely be interested in this paper because it addresses a timely and broadly relevant question for machine learning: how AI systems can better understand humans, rather than only improve task performance or predictive accuracy. The topic is relevant to researchers working on human-AI interaction, preference learning, AI agent and evaluation of generative models. Although the paper is more of a perspective piece than a novel technical contribution, its framing of Behavior AI may be useful for readers interested in human-centered machine learning and the broader society impact of AI systems.

**Broader Impact Concerns:**

The paper already discusses several important risks of Behavioral AI, including privacy invasion, manipulation, persuasion, deception, and misuse. However, given that the proposed goal is to build AI systems that better understand human preferences, emotions, goals, and intellectual states, the ethical implications are substantial and should be addressed more concretely. Also, I suggest that the authors expand the broader impact discussion with more actionable considerations, such as aspects including user control over personalization, privacy-preserving data collection, transparency about how the system safeguards against manipulative uses.

**Claims And Evidence:**

Yes

**Claims Explanation:**

The paper does not rely on a single technical result, but supports its perspective through a broad discussion of existing studies, concrete examples and arguments from multiple related fields. Its claims about the limitations of current AI systems are made convincing through examples of failures in preference understanding, intellectual emotional understanding. Its discussion of evaluation challenges is also well supported, especially the argument that observed behavior is often only a surface-level proxy for latent preferences, beliefs, goals and emotions. However, some broader claims, like the positioning of Behavior AI as a distinct new research field, are more conceptual than practically demonstrated and could be further strengthened by clearer definitions and more concrete evaluation framework.

**Requested Changes:**

1. The figures should be made more informative and self-contained. Currently, they do not add much conceptual information beyond the surrounding text. Since this is a perspective paper, the figures should help readers quickly understand the proposed Behavioral AI framework, rather than only serve as illustrative decorations.
2. Figure 2 should be revised for clarity. The current figure seems to describe a feedback loop among AI systems, users, and behavioral science, but the direction of influence and the meaning of each component are not sufficiently explicit. Clearer arrows, labels, and short annotations would make the figure easier to interpret.
3. The three dimensions of understanding: preferential, intellectual, and emotional understanding, should be more clearly operationalized. The paper explains them intuitively, but it would be helpful to clarify how these dimensions differ from each other, how they interact, and how they could be measured in practice.
4. The risk discussion should be made more actionable. The paper correctly identifies privacy, manipulation, persuasion, and misuse as risks, but it would be stronger if it proposed concrete safeguards, design principles, or governance considerations for Behavioral AI systems.
5. The paper could include a summary table mapping each dimension of human understanding to representative tasks, failure modes, possible evaluation methods and relevant existing fields. This would make the proposed research agenda easier to understand and would help clarify the novelty of the Behavioral AI framing.

---

> ### Author Response · Authors · 2026-07-17
> **Response to Reviewer zoWw**
>
> Thank you for your positive and constructive review. We respond to each requested change below.
>
> > _1. The figures should be made more informative and self-contained_
>
> Thank you for this suggestion. We've revised both figures to be more informative and self-contained. Figure 1 now contrasts a surface-level AI that predicts behavior (restocking the chips a person is trying to avoid) with an improved system that infers the intent beneath the behavior and supports the person's actual goal. It also includes more labeling in the figure and a more informative caption.
>
> > _2. Figure 2 should be revised for clarity. The current figure seems to describe a feedback loop among AI systems, users, and behavioral science, but the direction of influence and the meaning of each component are not sufficiently explicit. Clearer arrows, labels, and short annotations would make the figure easier to interpret._
>
> We apologize for the confusion! In hindsight, indeed this figure was confusing. We have revised Figure 2 to better clarify two key feedback loops: between an individual user and AI that understands us (producing mutual understanding) and between AI that understands us and the behavioral sciences (advancing our scientific understanding of people and conversely, offering insights into how to build better AI).
>
>
> > _3. The three dimensions of understanding: preferential, intellectual, and emotional understanding, should be more clearly operationalized. The paper explains them intuitively, but it would be helpful to clarify how these dimensions differ from each other, how they interact, and how they could be measured in practice._
>
> Great points. We have added a new table (Table 1) which aims to concretize the dimensions of understanding further. The beginning of Section 3 now states explicitly what we mean by understanding, and the revision uses two running examples to distinguish the dimensions (the smart pantry for preferential understanding and the trigonometry student for intellectual understanding), with the end of Section 2 discussing how the dimensions interact.
>
>
> > _4. The risk discussion should be made more actionable. The paper correctly identifies privacy, manipulation, persuasion, and misuse as risks, but it would be stronger if it proposed concrete safeguards, design principles, or governance considerations for Behavioral AI systems._
>
> We agree that the risks deserve a deeper treatment, and we've expanded the risks discussion in Section 5. It now directly discusses manipulation and persuasion tailored to inferred states, emotional dependence, privacy harms and exploitation by malicious actors, and misalignment, where systems optimized for engagement or approval turn their understanding against the people they model. It also lays out concrete safeguards and design principles, including user control and transparency over what a system infers, privacy-preserving data collection, limits on persuasive optimization, settings where systems should not infer at all, and the governance and consent questions these systems raise.
>
> > _5. The paper could include a summary table mapping each dimension of human understanding to representative tasks, failure modes, possible evaluation methods and relevant existing fields._
>
> Thank you for this suggestion. We've added a summary table (Table 1) mapping each dimension of understanding to representative tasks, failure modes, possible evaluation approaches, and relevant existing fields. We agree it makes the agenda easier to navigate. It also helped us clarify how the dimensions differ and is referenced from the dimensions discussion in Section 2.

---

### Author Response · Authors · 2026-07-17
**Response to all reviewers**

We thank all three reviewers for their careful and constructive reviews. We've revised the paper in response. The largest change, following Reviewer vRwg's suggestion, is a reframing so the paper no longer positions Behavioral AI as a new field, but instead as a grand challenge (building algorithms that understand people) that many existing fields are already working toward, albeit often in isolation. Beyond the reframing, the major changes are:
- A new discussion of related fields in Section 1, with a new table summarizing each field's contributions and open challenges
- A new table mapping each dimension of understanding to representative tasks, failure modes, and candidate evaluations
- An expanded risks discussion in Section 5, covering manipulation, emotional dependence, privacy harms, misalignment, and concrete safeguards
- New material on personalization (Section 4), multimodal signals (Section 4), and human-human baselines (Section 3.2)
- Revised figures and a tightened pass over prose.

We respond to each point individually below.